# Longitudinal detection of somatic mutations in saliva and plasma for the surveillance of oral squamous cell carcinomas

Ying Cui[1,2]☯, Hae-Suk Kim[3]☯, Eunae Sandra Cho[2,4], Dawool Han[2,4], Jung Ah Park[3], Ju Yeong Park[3], Woong Nam[1], Hyung Jun Kim[1], In-Ho Cha[1], Yong Hoon Cha [1,2]*

1 Department of Oral and Maxillofacial Surgery, Yonsei University College of Dentistry, Seoul, Republic of Korea, 2 Oral Cancer Research Institute, Yonsei University College of Dentistry, Seoul, Republic of Korea, 3 Theragen Bio Co., Ltd, Seongnam-si, Republic of Korea, 4 Department of Oral Pathology, Yonsei University College of Dentistry, Seoul, Republic of Korea

☯ These authors contributed equally to this work.

* omfscha@yuhs.ac

**Data Availability Statement:** All relevant data are within the manuscript and its Supporting Information files. Next generation sequencing data were uploaded on NCBI SRA with accession

## Abstract

### Purposes

Although clinical and radiological examinations can be used to diagnose oral cancer, and surgical pathology remains the gold standard, these conventional methods have limitations. We evaluated the feasibility of longitudinal next-generation sequencing-based liquid biopsy for oral squamous cell carcinoma surveillance.

### Materials and methods

Eleven patients were enrolled, and plasma and saliva were collected before, and 1, 3, and 6 months after surgery. Tumor-specific mutations were selected using paired, whole-exome analyses of tumor tissues and whole blood. Genes frequently mutated in head and neck cancer were identified using the Cancer Genome Atlas (TCGA) and Catalogue of Somatic Mutations in Cancer (COSMIC) databases to design targeted deep sequencing panels.

### Results

In five of the six patients with recurrent cancer, circulating tumor DNA (ctDNA) was detected earlier with liquid biopsy than with conventional monitoring techniques. Moreover, patients without recurrence exhibited decreased ctDNA allele frequency post-treatment.

### Conclusions

Longitudinal liquid biopsy of plasma and saliva may be feasible for detecting somatic mutations associated with oral squamous cell carcinomas. It might be attributable to determine early tumor recurrence through genetic analysis of ctDNA.

number PRJNA718901 (https://www.ncbi.nlm.nih.gov/bioproject/PRJNA718901/).

**Funding:** This work was supported by the National Research Foundation of Korea (NRF) grant funded by the Korean Government (MSIT) (No. 2018R1C1B600445813). The funder provided support in the form of salaries for author YC, but did not have any additional role in the study design, data collection and analysis, decision to publish, or preparation of the manuscript. Theragen Bio., Ltd. provided support in the form of salaries for authors HSK, JAP and JYP, but did not have any additional role in the study design, data collection and analysis, decision to publish, or preparation of the manuscript. There are no patents, products in development, or marketed products to declare.

**Competing interests:** HSK, JYP, and JAP, who are employees of Theragen Bio, solely contribute their scientific expertise and knowledge to conduct next-generation sequencing and bioinformatics analysis for this study. They do not have any competing interests. Also, I would like to declare that Theragen Bio did not provide any funding for this study and does not alter my adherence to PLOS ONE policies on sharing data and materials.

## Introduction

The current clinical methods for cancer diagnosis include clinical examinations, surgical biopsy, and imaging modalities such as computed tomography (CT), magnetic resonance imaging (MRI), and positron emission tomography (PET). Among them, surgical biopsy and subsequent pathologic diagnosis are the gold standard for oral cancer diagnosis, even though these procedures have several limitations. Biopsies are invasive, time-consuming procedures that are difficult to repeat [1–3]. Furthermore, a biopsy does not reflect the spatiotemporal heterogeneity of a solid tumor because it targets only a single tumor site at a specific time point [4]. Unfortunately, owing to the paucity of disease-specific biomarkers for oral cancer, clinicians tend to rely only on conventional diagnostic tools [5, 6].

Although radiological examinations are useful in oral cancer diagnosis, they also have some limitations. Radiological examinations are challenging to use while observing orofacial lesions, not only due to metal artifacts from prostheses, dental implants, fixation plates, and screws, but also owing to post-operation fibrosis and inflammation during all periods of treatment [7]. In this regard, early surveillance using PET/CT may not provide accurate results until 12 m after initial treatment [8].

Perioperative surveillance is closely related to tumor recurrence and overall survival. Several clinical factors, such as nodal status, invasion depth, and surgical margin, have already been identified as prognosticators in oral cancer [6, 9, 10]. As early recurrence tends to significantly decrease overall survival rates in oral cancer [11, 12], early detection of tumor recurrence is crucial to improve the survival rate.

Recently, minimal residual disease (MRD) monitoring has emerged as an important technique in tumor surveillance. MRD refers to a microscopic tumor that persists during or after treatment and cannot be diagnosed by conventional clinical or imaging methods [13, 14]. In hematologic cancers, MRD detection and monitoring are well-established and widely used [15, 16]. However, their application for solid tumors remains challenging owing to difficulties in sampling the low concentrations of circulating cells and other factors that cancer cells secrete into the bloodstream [14, 15]. Hence, serial quantitative methods have been unable to fulfill all the requirements of oral cancer diagnosis and surveillance.

The recently developed liquid biopsy technique can be an alternative diagnostic modality for oral cancer [17–19]. However, prospective studies examining the use of complex mutation, subset-based liquid biopsies for the quantitative serial detection of tumor recurrence in oral cancer before and after treatment are unavailable. Hence, this study aimed to examine the feasibility of using serial liquid biopsy in detecting MRD in patients with oral cancer.

## Materials and methods

### Samples

We studied tumor tissues and liquid samples from 11 patients with oral cancer. The study was approved by the institutional review board of Yonsei University Dental Hospital, Seoul, Republic of Korea (Approval No. 2018–0061), and performed in accordance with the Declaration of Helsinki. Individual written informed consent was obtained from all participating patients at enrolment. The participants were limited to those who underwent surgery among patients histologically diagnosed with oral squamous cell carcinoma at the research institute between April and July 2019, and all races were asian. Oropharyngeal cancer, salivary gland cancer, mucosal melanoma and sarcoma were excluded, and children and adolescents under the age of 18 and illiteracy were excluded. After the start of the study, we participated in the collection of liquid samples until 6 months after surgery, and clinical recurrence was

monitored until 18 months after surgery. All participants voluntarily signed up after hearing the explanation of the study in an independent space before surgery, and sampling was also performed in a blood collection room and in an enclosed independent space. All participants were confirmed as oral squamous cell carcinoma through incisional biopsy by pathologists. Also intraoperative fresh frozen biopsies were performed until acquiring free margin.

Fresh tumor tissues were collected during the operation following macro dissection to ensure that neoplastic cellularity was over 20% and were then snap-frozen in liquid nitrogen. In case fresh tumor tissue was insufficient (patients LB-005, LB-007), formalin-fixed, paraffin-embedded (FFPE) tissues were collected through laser captured microdissection (Leica LMD 6500, Leica LMD membrane slide), achieving a tumor cellularity of over 80%. Genomic DNA was isolated from tumor tissue using the QIAamp DNA micro kit (Qiagen, Hilden, Germany) following the manufacturer's instructions. Three milliliters of paired whole blood were used to isolate genomic DNA from each patient using Intron G-Dex$^{TM}$IIb (Intron, Seongnam-Si, Korea). Saliva was collected using sterilized 50 ml tubes (Eppendorf, Germany) after oral rinsing with sterilized saline. In total, 5–15 ml of saliva was collected each time and snap-frozen for storage. Chewing paraffin (Ivoclar Vivadent, USA) was used for patients who underwent radiotherapy (RT) or concurrent chemoradiotherapy (CCRT) if they complained of xerostomia. The saliva was centrifuged at 1,600 $g$ for 10 min at 4˚C, and the clear supernatant was transferred to a new 50 ml tube and centrifuged at 1,600 $g$ for 5 min at 4˚C. Ten milliliters of whole blood were collected in Cell-Free DNA blood collection tube (BCT) (Streck, USA) for harvesting circulating tumor DNA (ctDNA). Blood samples were centrifuged at 1,600 $g$ for 10 min at 22˚C. Plasma was then separated by centrifugation at 1,600 $g$ for 10 min at 4˚C to remove cell debris; subsequently, 1 ml aliquots were placed in Eppendorf tubes and stored at −80˚C before extraction. Cell-free DNA (cfDNA) was isolated from both saliva and whole blood using MagMax (Thermo Fisher Scientific, USA) following the manufacturer's instructions. A mean of 4 ml of plasma was used for cfDNA isolation. All DNA samples were stored at –80˚C and quantified using the Qubit fluorometer (Invitrogen, USA), and the size distribution was evaluated using the 2200 Tapestation (Agilent Technologies, USA). The schematic workflow of sample collection, DNA isolation, and sequencing is depicted in Fig 1. Both saliva and plasma were longitudinally collected at four time points for the isolation of cfDNA: T1, immediately before surgery; T2, 1 m after surgery, T3, 3 m after surgery, and T4, 6 m after surgery.

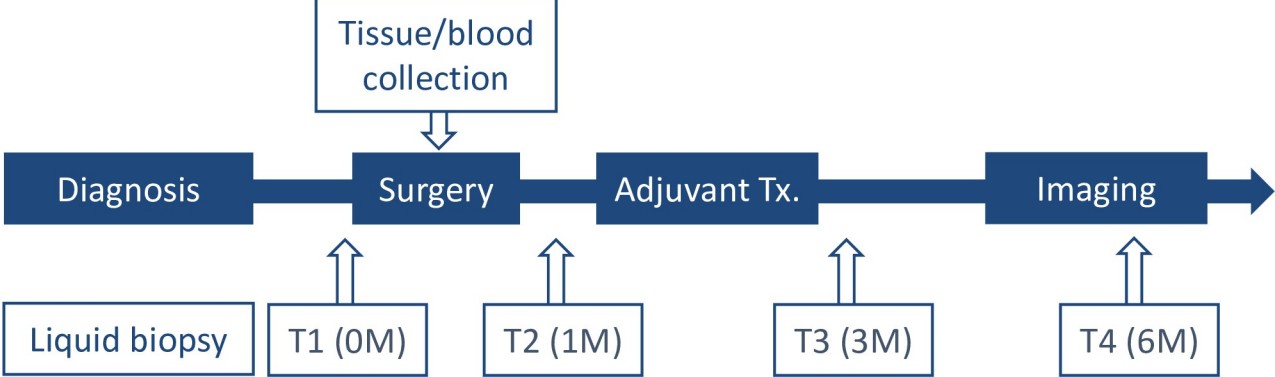

**Fig 1. Schematic diagram of the longitudinal study design.** Both plasma and saliva were collected for evaluation using liquid biopsy. For example, T2 samples were used to estimate the therapeutic effect of surgery based on T1 sampling data, and T3 samples were used to evaluate the efficiency of adjuvant therapies based on sampling T2 sampling data.

## Next-generation sequencing

Whole exome sequencing (WES) was conducted using tumor tissue and paired whole-blood samples to identify tumor-specific somatic mutations. Targeted deep sequencing for cell-free DNA from liquid samples was performed to detect ctDNA. Briefly, genomic DNA from each sample was fragmented by acoustic shearing using a Covaris S2 instrument (Covaris, USA), but cfDNAs were skipped in this step. Fragments of 150–300 bp were ligated to the Illumina adapters and polymerase chain reaction (PCR)-amplified. The samples were concentrated and hybridized with RNA probes, and the SureSelect XT Human All Exon V5 Capture library for WES or oral cancer-specific customized gene panel was used for targeted deep sequencing. To design the panel, tumor-specific, high impact somatic mutations were chosen and compared to germline mutations from the paired whole blood sample. In addition to patient-driven mutations, mutations frequent in head and neck cancer were chosen using both The Cancer Genome Atlas (TCGA) from the Genomic Data Commons (GDC) data portal (National Cancer Institute), and the Catalogue of Somatic Mutations in Cancer (COSMIC) Mutant Census v87.

After hybridization, the captured targets were pulled down by biotinylated probe/target hybrids using streptavidin-coated magnetic beads (Dynabeads My One Streptavidin T1; Life Technologies Ltd.) and buffers. The selected regions were then PCR-amplified using Illumina PCR primers. Libraries were quantified using the Agilent 2100 Bioanalyzer (Agilent Technologies), and KAPA Library Quantification Kit (KK4824, Kapa Biosystems). The resulting purified libraries were applied to an Illumina flow cell for cluster generation and sequenced using 100 bp paired-end reads on an Illumina NovaSeq6000 sequencer using the manufacturer's protocols.

## Sequencing quality control

The quality of the reads was checked using fastQC (v.0.11.8), which revealed the basic quality in terms of the sequence quality score, GC content, N content, length distribution, and duplication level. After checking the read quality, low-quality bases below Q20 were trimmed using Cutadapt (v.2.5) [20].

## Sequencing alignment, variant call, and annotation

High-quality reads were then aligned to the human reference genome, hg19, using the Burrows-Wheeler Aligner (BWA) (v.0.7.17) [21]. Subsequently, the duplicated reads without UMI were removed using MarkDuplicates.jar in PicardTools (v.2.20.7). For deduplication reads with UMI, the fgbio (v.1.0.0) packages was used. Intervals showing mismatch were examined for more accurate realignment using the Realigner Target Creator tool in the Genome Analysis toolkit (GATK) (v.4.1.3) [22], and 'Mills-and-1000G-gold.standard-INDELs.hg19' was used as the known indel set. The base quality score recalibration (BQSR) process was then performed to adjust the quality score using the Base Recalibrator tool in GATK. For the realigned and recalibrated reads, variants were called using the Mutect2 tool in GATK. All variants were then annotated using SnpEff (v.4.3.1) [23], based on predictions for the deleterious or clinical effect of the variants using dbNSFP [24], COSMIC [25], and ClinVar [26]. Allele frequencies were obtained from the 1000 Genomes [27], ESP6500, and ExAC databases [28]. Mutation allele frequencies over 0.1% indicated the presence of ctDNA in liquid samples. In contrast, high-frequency mutations (frequency over 7.0%) collected from open-source data were considered noise. Additionally, mutations collected from TCGA and COSMIC were considered true mutations when they were detected in pre-treatment liquid samples. Also, all synonymous

variants with damaging impact (Low or MODIFIER) as annotated by Ensembl VEP were filtered out.

For generating oncoplot to display somatic landscape between patients, R package maftools (v.0.9.30) was used. The Customized R code was used to generate a swimmer plot to track the occurrence of mutations in patients over time. The variant concordance was demonstrated by customized Python code.

## Results

### Clinical features and mutations in primary tumors

A total of 11 patients were enrolled in this prospective, longitudinal study. All patients were diagnosed with squamous cell carcinoma before surgery by an experienced oral and maxillofacial pathologist. Their average age was 64 and seven of them were female (63.6%). All samples were collected from the oral cavity. Except for individual tongue and cheek samples, nine samples were from the gingiva (five from the maxilla and four from the mandible). Four patients (36.4%) had early (stage II) disease, and the remaining seven patients (63.6%) had advanced (stage III or IV) disease. Among those with advanced disease, three patients had a history of recurrent tumors. In particular, patient LB-002 had five tumor recurrences only in the oral cavity, with a history of three surgical and two RT interventions. Furthermore, patient LB-011 had a history of definitive RT on the right tongue 10 years before this study. In one patient (LB-007), the tumor had recurred at the same site within eight months of initial surgical treatment before this study (Table 1).

To begin this study, we attempted to identify tumor-only somatic mutations in every patient through WES and compared these to germline mutations from whole blood. *TP53* mutations were a driver event in six patients (55%), while *CASP8*, *AJUBA*, *CDKN2A*, and *NOTCH1* mutations were also frequent (Fig 2). *PIC3CA* mutations were not observed in the primary tumors in our cohort.

### Variant selection for panel design

In addition to tumor-specific mutations from tissue samples, frequent somatic mutations were searched for in TCGA and COSMIC to produce a panel optimized for head and neck

**Table 1. Summarized clinical information from patients enrolled in the longitudinal liquid biopsy study for the detection of oral squamous cell carcinomas.**

| Patient no. | Age | Sex | Primary site | TNM (stage) |
|---|---|---|---|---|
| LB-001 | 40 | F | Tongue, left | T3N0M0 (III) |
| LB-002* | 67 | F | Mx. gingiva, left | T2N0M0 (II) |
| LB-003 | 52 | M | Mn. gingiva, left | T4bN0M0 (IVb) |
| LB-004 | 73 | F | Mx. gingiva, anterior | T2N0M0 (II) |
| LB-005 | 77 | F | RMT, right | T4aN0M0 (IVa) |
| LB-006 | 63 | M | Mx. gingiva, right | T4aN0M0 (IVa) |
| LB-007* | 57 | F | Cheek, right | T4bN0M0 (IVb) |
| LB-008 | 75 | M | Mx. gingiva, right | T4aN0M0 (IVa) |
| LB-009 | 84 | F | Mx. gingiva, right | T2N0M0 (II) |
| LB-010 | 53 | F | Mn. gingiva, right | T3N0M0 (III) |
| LB-011* | 62 | M | Mn. gingiva, left | T2N0M0 (II) |

Asterisk (*) indicates recurrent tumors at the beginning of the study. The TNM stage was decided based on the 8[th] edition of the AJCC.

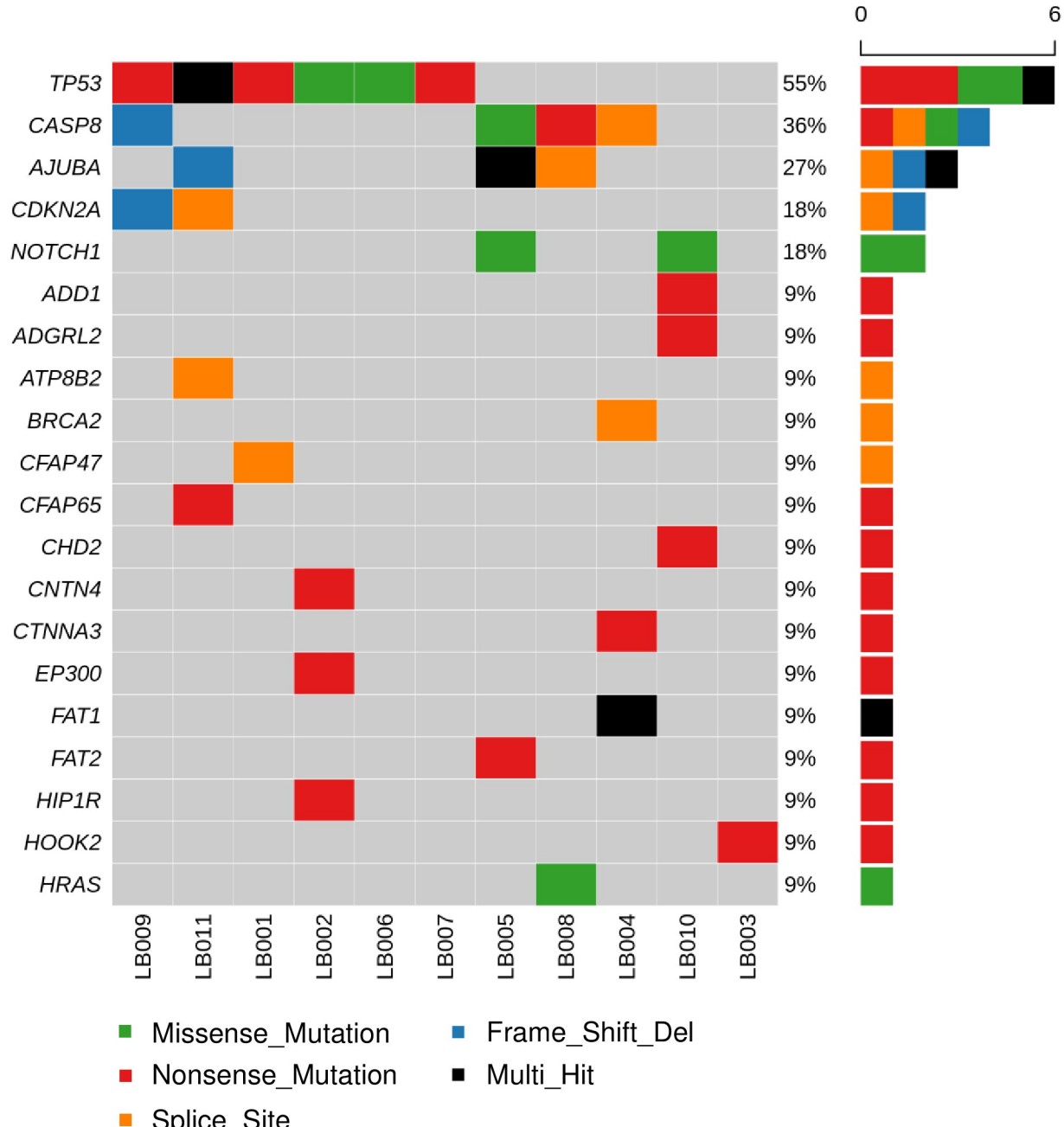

**Fig 2. Oncoplot of the top 20 genes analyzed by whole-exome sequencing from the index tumors of 11 patients.** Sequencing depth was 200 x for the index tumor, and 100 x for whole blood.

squamous cell carcinoma (HNSCC), involving oral cancer (S1 and S2 Tables). In total, 308 mutations were selected from a total of 508 TCGA HNSCC samples. *TP53* was the most frequently mutated gene overall with various mutations (531/834, 63.7%). The most frequent independent mutation in TCGA was *PIK3CA* c.1633G > A (p.Glu545Lys) with an overall frequency of 5%. *TP53* c.524G > A (p.Arg175H) mutation was the most frequent mutation among the *TP53* mutations, with a 2% frequency overall. Among the 185 mutations summarized from COSMIC, *TP53* showed the most frequent mutation count (588/3613, 16.3%),

followed by *CDKN2A* and *PIK3CA*. Interestingly, although *TP53* was the most mutated gene in both TCGA and COSMIC, the frequency of mutation differed in each set. Furthermore, previously reported driver events were considered for designing the panel involving *FAT1*, *AJUBA*, *NOTCH1*, *HRAS*, *NRAS*, *FBXW7*, *KMT2D*, and *NSD1* [17, 29]. All these given mutations were summarized to produce the panel with the most frequent genes (S3 Table).

## Longitudinal liquid biopsy for oral cancer surveillance

Longitudinal liquid biopsies using saliva and plasma were performed on 11 patients. As nine samples (five plasma and four saliva samples) were missed and two saliva samples did not pass the DNA quality control, a total of 77 liquid samples underwent deep sequencing. Targeted deep sequencing was performed to detect the ctDNA based on the previously designed panel.

Of the 11 patients involved in the study, clinical recurrence was confirmed at the 18 m follow-up after initial treatment in six patients. Four of them had an advanced stage (stage III, IV), and four had close or positive surgical margins (close 1/6, 16.7%, positive 3/6, 50.0%), while the rest had a free margin (2/6, 33.3%). In contrast, 4 out of 5 (80%) patients without recurrence had a free margin, and the remaining one had close hard-tissue margin (1/5, 20%). Four of six patients (66.7%) with recurrent cancer received RT (1 patient) or CCRT (3 patients) for adjuvant therapy, while only one patient underwent adjuvant RT in the non-recurrent group (1/5, 20%) (Table 2, Fig 3A and 3B).

In this study, we successfully detected the presence of ctDNA through a combination of liquid samples mainly dependent on saliva from five patients with recurrent cancer (Fig 3C, Table 2, S4 Table). Only one patient (LB-003), who showed clinical recurrence at 12 m, did not show the presence of ctDNA until 6 m after initial treatment. Notably, the recurring lesion was observed in the left temporal space, while the primary tumor was found on the ipsilateral gingiva of the mandible. The mean time difference between first longitudinal ctDNA detection and clinical recurrence diagnosis was 4.4 m (mean 2.4 m versus 6.8 m) in the five patients showing recurrence. Most clinical recurrences were found on regular imaging studies (CT, MRI) and clinical examinations, such as naked-eye inspection or palpation (Table 2). And the mean allele frequency of mutations detected in saliva of each patient was relatively increased in recurred patients at both T2 (1 month after initial treatment) and T3 (3 month after initial

**Table 2. Summary of longitudinal ctDNA detection results and clinical recurrence.**

| Patient no. | ctDNA detection (months post-surgery) | | | | Clinical recurrence | Recurrence recognition time (months) | Adjuvant therapy | Recurrence detection method |
|---|---|---|---|---|---|---|---|---|
| | 0 | 1 | 3 | 6 | | | | |
| LB-001 | O | NA | O | O | Recur | 6 | RT | Clinic, CT |
| LB-002 | O | O | O | O | Recur | 8.5 | - | Clinic, pathology |
| LB-003 | O | - | - | - | Recur | 12 | CCRT | Clinic, MRI |
| LB-004 | O | - | - | - | - | - | - | - |
| LB-005 | O | - | NA | - | - | - | - | - |
| LB-006 | O | - | - | O | Recur | 6 | CCRT | MRI |
| LB-007 | O | O | NA | NA | Recur | 7.5 | CCRT | MRI, CT |
| LB-008 | O | O | O | O | Recur | 6 | - | MRI, CT |
| LB-009 | O | O | - | - | - | - | - | - |
| LB-010 | O | - | - | - | - | - | RT | - |
| LB-011 | O | NA | - | - | - | - | - | - |

In five out of six patients, locoregional recurrences were detected earlier with liquid biopsy than with conventional methods. NA: not available.

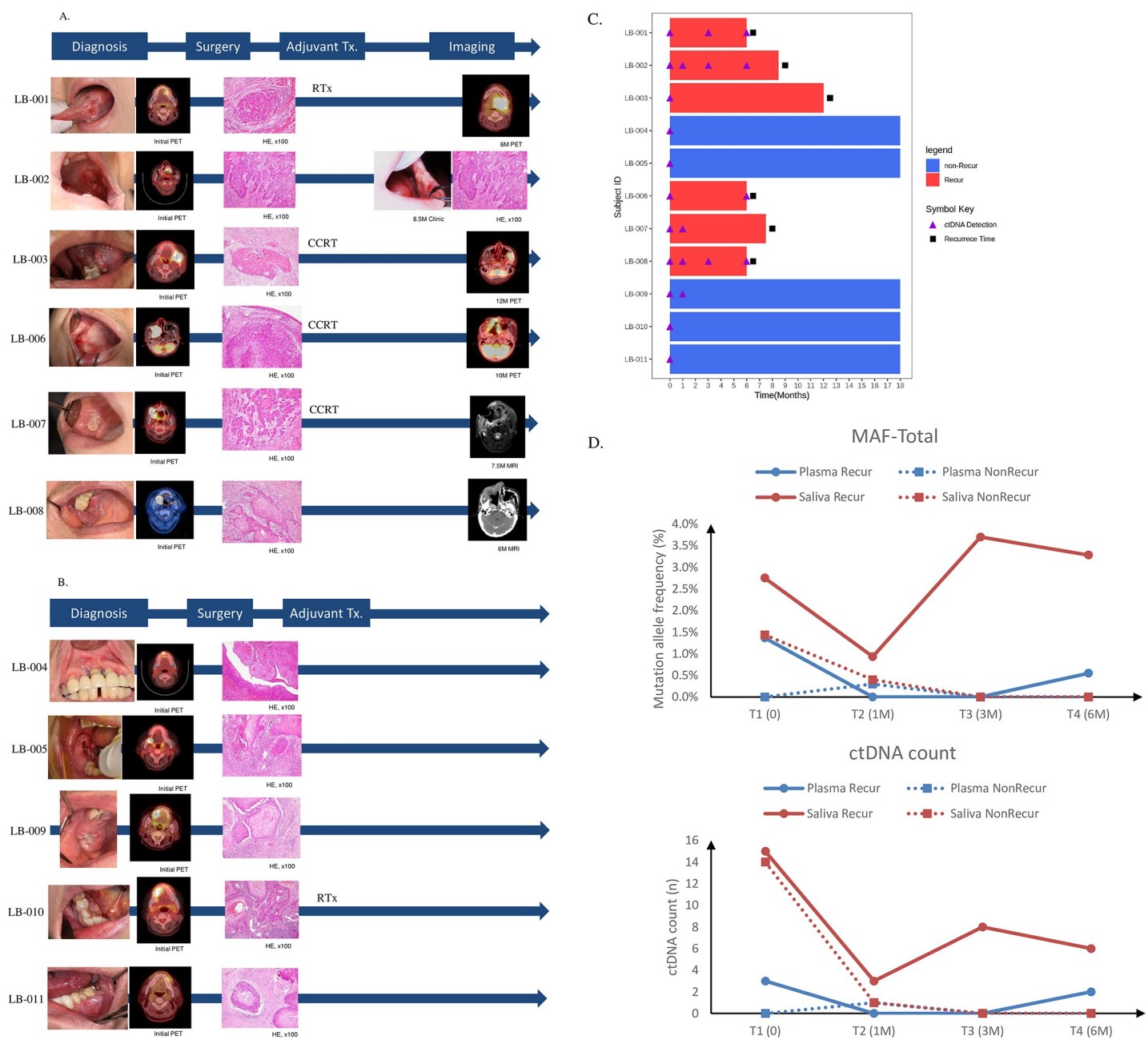

**Fig 3. Clinical progress of recurrence and non recurrence group.** Mean allele frequency (MAF) and ctDNA count differences between plasma and saliva based on liquid biopsy examinations. A and B. Clinical monitoring of both recurrence group (A) and non recurrence group (B), C. A swimmer plot to track the occurrence of mutations in patients over time. The purple triangle indicates the detection of ctDNA. D. MAF of both plasma and saliva samples, Sum of ctDNA counts in both plasma and saliva.

treatment) compared with the non-recurred (mean AF, 0.05% for non-recurred versus 1.58% for recurred; median AF, 0% for non-recurred versus 0.6% for recurred; Wilcoxon sighed rank test, *p* 0.03) (S1 Fig). Based on this result the presence of ctDNA in liquid biopsy indicated a substantial increased possibility of recurrence. Earlier ctDNA detection indicated the need for close clinical monitoring such as interdisciplinarity approach with ctDNA testing at successive time point.

**Table 3. Summary of surveillance of oral squamous cell carcinoma recurrence by longitudinal liquid biopsy.**

| Time point (months post-surgery) | Recurrence | Plasma (%) | Saliva (%) | Saliva or Plasma (%) |
|---|---|---|---|---|
| 0 | - | 3/11 (27) | 10/11 (91) | 11/11 (100) |
| 1 | No | 1/4 (25) | 1/4 (25) | 1/4 (25) |
| | Yes | 0/5 (0) | 3/6 (50) | 3/6 (50) |
| 3 | No | 0/4 (0) | 0/4 (0) | 0/4 (0) |
| | Yes | 0/5 (0) | 3/4 (75) | 3/6 (50) |
| 6 | No | 0/5 (0) | 0/5 (0) | 0/5 (0) |
| | Yes | 2/5 (40) | 2/5 (40) | 4/5 (80) |

Saliva samples were better than plasma samples for oral cancer surveillance. Some patients with recurrence exhibited low ctDNA levels in saliva at one and three months post surgery.

Among the patients without recurrence, patient LB-009 showed the *TP53* c.818G > A single nucleotide variant (SNV) in both plasma and saliva initially and at 1 m after surgery which was not observed on the index tumor. However, those ctDNA SNV disappeared after 3 m even though the patient did not receive any adjuvant therapy. Intriguingly, the surgical margin of this patient was positive only in maxilla hard tissue but free in soft tissue. This patient suffered from oroantral fistula and necrosis on the surgical site after surgery.

The efficiency of liquid biopsy for oral cancer was 100% before initial treatment when combining plasma and saliva, while the latter showed higher efficiency alone. Although ctDNA detection with saliva missed only one patient (LB-006) who had stage IV maxilla gingiva cancer, the presence of plasma ctDNA in the same patient compensated for this. When patients were grouped following clinical recurrence, ctDNA detection efficiency was relatively low at T2 (1 m after initial treatment) (Table 3). Considering that a few liquid samples were not collected, ctDNA in saliva was detected with a greater sensitivity than ctDNA in plasma immediately after surgery. In particular, sequencing could not be performed on the two T3 saliva samples from the recurrence group as they did not meet the DNA quality control criteria. Although the detection rate of ctDNA in saliva decreased in the recurrence group at T4 when plasma ctDNA started being detected, the overall tumor monitoring efficiency was higher when saliva and plasma were both used.

Intriguingly, mean allele frequency (MAF) and ctDNA count through targeted deep sequencing in plasma and saliva samples provided quantitative information. Saliva constantly showed a much higher MAF and sum of ctDNA counts than plasma (Fig 3D). Saliva also exhibited that the local residual tumor was absent in the group that did not have cancer recurrence as time progressed. Although plasma had a similar expression pattern as saliva, saliva had much higher quantitative values in both the MAF and ctDNA count.

## Concordance of variants between liquid biopsy and index tumor

The concordance of targeted deep sequencing results was estimated in samples from 11 patients using cfDNA and genomic DNA from primary tumor tissues before initial treatment. The concordance rate between salivary cfDNA and tumor tissue DNA was 72.7% (8/11) among all patients while the rate for plasma cfDNA was 9.1% (1/11). In salivary cfDNA, two patients showed complete concordance and six patients showed partial concordance (Fig 4A). The concordance for each gene is depicted in Fig 4B. Although two patients (LB-001,009) did not exhibit concordance between tumor tissue and liquid samples, *TP53* mutations that were not found on WES of the index tumor were identified by deep sequencing of liquid samples.

## Discussion

This study was designed to develop a workflow for the detection of SNVs in cfDNA from the plasma and saliva of patients with oral cancer using an identical sequencing panel and to evaluate its feasibility in longitudinally evaluating liquid samples of 11 patients with oral cancer. This method was based on targeted deep sequencing of a panel composed of frequently mutated genes in HNSCC. This panel-based, multiplex next-generation sequencing approach provides several advantages for liquid biopsy. High-resolution data can be acquired through multiplex analysis using the specialized HNSCC panel. It can also help to overcome weakness in the selection of unique mutations from index tumors that cannot fully cover the evolution tracing and dynamic changes in the mutational spectrum and intra-tumor genetic heterogeneity [4, 14, 30, 31].

Overall, we explored the relevance of liquid biopsy to monitor the clinical recurrence of oral cancer using a designed panel at deep coverage. As a result, we successfully detected the presence of ctDNA in both plasma and saliva of patients with oral cancer, and the latter exhibited much higher resolution. The presence of a primary tumor was 100% confirmed before initial treatment by liquid biopsy in a previous report [17]. Quantitatively, increased SNVs from ctDNA were observed approximately three months after surgery mainly in saliva samples and not in plasma samples, which suggests the possibility of locoregional recurrence. The emergence of SNVs of ctDNA in liquid samples was detected approximately four months earlier than clinical recurrence by conventional clinical methods, and until 6 m after surgery, we continued sometimes-blinded radiological examinations as post-operation adjuvant therapy.

As genetic variation in cancer undergoes dynamic changes from emergence to evolution that overcome therapies, intra-tumor and inter-tumor heterogeneity tends to increase [4, 32]. This pattern implies that a few driver mutations from index tumors cannot cover serial tracing of tumor recurrence. Thus, frequent-mutation selection from both TCGA and COSMIC, in addition to index tumor sequencing, can be an alternative to serial liquid biopsy.

Previous reports used HPV-DNA presence in plasma to increase detection efficiency in HPV-positive, oropharyngeal cancer [17]. However, this method is difficult to apply in oral cancer because most oral cavity cancers (e.g., gum, palate, tongue, and cheek) are HPV-negative [33]. Therefore, ctDNA or copy number aberrations (CNA) in local or systemic fluids may be preferable for liquid biopsy in patients with oral cancer. Recent reports have shown advantages to low-coverage, whole genome sequencing in HNSCC patients, while the ctDNA detection method is still too obscure for application in a patient cohort [18]. We also successfully detected the evidence of tumor recurrence with ctDNA in the plasma of patients while the probability of detection was higher in saliva than in plasma. Although it cannot be concluded whether whole genome sequencing or targeted sequencing is clinically more effective in HNSCC, it is important to determine the timing of sampling as liquid biopsy is clinically most useful for monitoring early recurrence after initial treatment than pretreatment or metastatic setting [19].

Ideally, driver-mutation selection from index tumors and subsequent, large-panel-based, serially targeted deep sequencing for ctDNA and low-coverage, whole genome sequencing for CNAs, along with digital droplet PCR for validation, are all important research methods for tracing tumor recurrence and evolution. These techniques remain useful even with restrictions on time, funds, and cfDNA quality. The other unexpected clinical obstacle in collecting saliva for the cfDNA study was xerostomia (severely decreased saliva secretion) in patients that underwent RT or CCRT. As saliva showed better efficiency in locoregional recurrence monitoring in this study, overcoming low quantity and quality of salivary cfDNA would be the main hurdle in future studies. Another limitation is the small sample size of this patient cohort,

A.

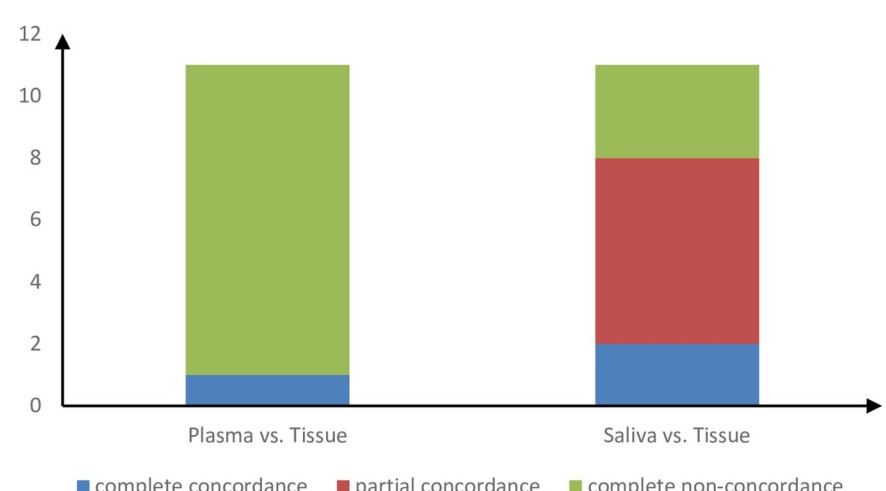

B.

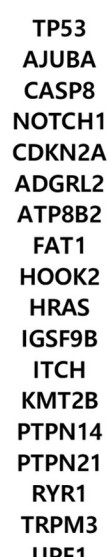
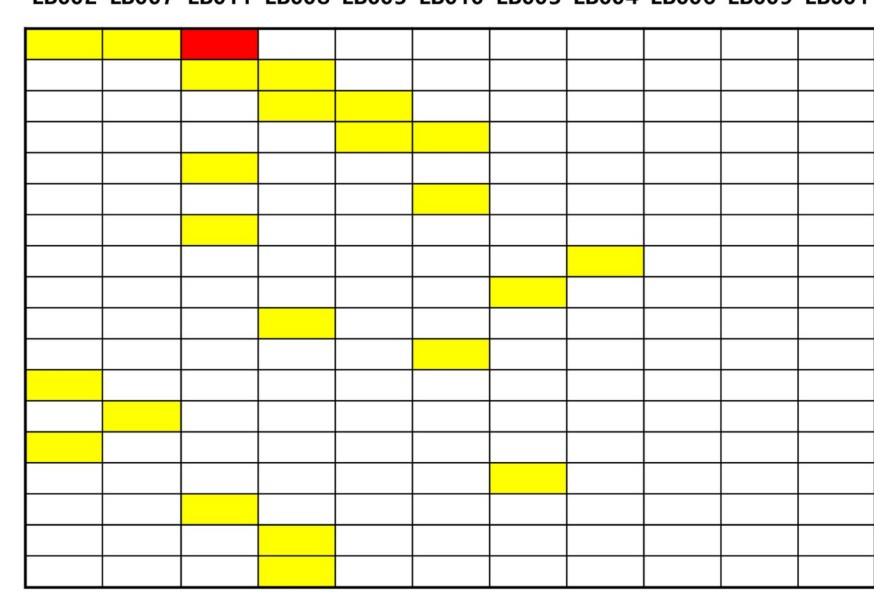

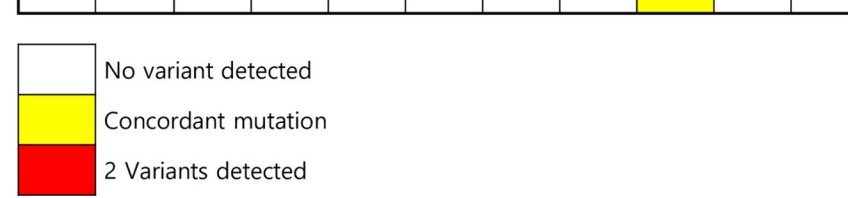

**Fig 4. Concordance between cfDNA and index tumors.** A. Summarized concordance between plasma and saliva samples, B. Summarized gene-based concordance.

hampering us to correlate our findings with the clinical outcome. When calculated through G-power [34], it is recommended that the minimum sample size for repeated measurements is 23 or more. However, due to the constraints of cost and time, 11 patients were enrolled in the study. Since it is difficult to represent the entire oral cancer patient population, a study with larger sample size will be needed in the future.

In conclusion, we examined the clinical feasibility of longitudinal liquid biopsies for the analysis of somatic mutations in cfDNA from both plasma and saliva samples of patients with oral cancer. Longitudinal liquid biopsy of plasma and saliva may be feasible for detecting somatic mutations associated with oral squamous cell carcinomas. Future research should focus not only on clinical protocol establishment for the surveillance of locoregional and metastatic disease, but also on designing novel clinical trials to provide practical benefits for these patients.

## Supporting information

**S1 Fig. Statistical analysis of earlier variant detection from recurrent patients.**
(TIF)

**S1 Table. Frequent somatic mutations found on head and neck squamous cell carcinoma of TCGA dataset.**
(XLSX)

**S2 Table. Frequent somatic mutations found on head and neck squamous cell carcinoma of COSMIC dataset.**
(XLSX)

**S3 Table. Gene list of customized targeted sequencing panel.**
(XLSX)

**S4 Table. The results of ctDNA detection in both plasma and saliva.**
(XLSX)

## Author Contributions

**Conceptualization:** Hae-Suk Kim, Yong Hoon Cha.

**Data curation:** Ying Cui, Hae-Suk Kim, Yong Hoon Cha.

**Formal analysis:** Ying Cui, Hae-Suk Kim, Jung Ah Park, Ju Yeong Park, Yong Hoon Cha.

**Funding acquisition:** Yong Hoon Cha.

**Investigation:** Ying Cui, Hae-Suk Kim, Yong Hoon Cha.

**Methodology:** Ying Cui, Hae-Suk Kim, Yong Hoon Cha.

**Project administration:** Hae-Suk Kim, Yong Hoon Cha.

**Resources:** Ying Cui, Hae-Suk Kim, Eunae Sandra Cho, Dawool Han, Woong Nam, Hyung Jun Kim, In-Ho Cha, Yong Hoon Cha.

**Software:** Jung Ah Park, Ju Yeong Park, Yong Hoon Cha.

**Supervision:** Hae-Suk Kim, Yong Hoon Cha.

**Validation:** Ying Cui, Hae-Suk Kim, Yong Hoon Cha.

**Visualization:** Ying Cui, Yong Hoon Cha.

**Writing – original draft:** Ying Cui, Hae-Suk Kim, Yong Hoon Cha.

**Writing – review & editing:** Ying Cui, Hae-Suk Kim, Yong Hoon Cha.

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
