## [Decision Letter · Decision Letter 0]

24 Mar 2021

PONE-D-21-08145

Longitudinal detection of somatic mutations in saliva and plasma for the surveillance of oral squamous cell carcinomas

PLOS ONE

Dear Dr. Cha Yong Hoon,

Thank you for submitting your manuscript to PLOS ONE. After careful consideration, we feel that it has merit but does not fully meet PLOS ONE’s publication criteria as it currently stands. Therefore, we invite you to submit a revised version of the manuscript that addresses the points raised during the review process.

We look forward to receiving your revised manuscript.

Kind regards,

Zhaoqiang Zhang

Academic Editor

PLOS ONE

Journal Requirements:

We note that one or more of the authors are employed by a commercial company: Theragen Bio Co., Ltd.

2.1. Please provide an amended Funding Statement declaring this commercial affiliation, as well as a statement regarding the Role of Funders in your study. If the funding organization did not play a role in the study design, data collection and analysis, decision to publish, or preparation of the manuscript and only provided financial support in the form of authors' salaries and/or research materials, please review your statements relating to the author contributions, and ensure you have specifically and accurately indicated the role(s) that these authors had in your study. You can update author roles in the Author Contributions section of the online submission form.

2.2. Please also provide an updated Competing Interests Statement declaring this commercial affiliation along with any other relevant declarations relating to employment, consultancy, patents, products in development, or marketed products, etc.  

3. We note that you are reporting an analysis of a microarray, next-generation sequencing, or deep sequencing data set. PLOS requires that authors comply with field-specific standards for preparation, recording, and deposition of data in repositories appropriate to their field. Please upload these data to a stable, public repository (such as ArrayExpress, Gene Expression Omnibus (GEO), DNA Data Bank of Japan (DDBJ), NCBI GenBank, NCBI Sequence Read Archive, or EMBL Nucleotide Sequence Database (ENA)). In your revised cover letter, please provide the relevant accession numbers that may be used to access these data. For a full list of recommended repositories, see http://journals.plos.org/plosone/s/data-availability#loc-omics or http://journals.plos.org/plosone/s/data-availability#loc-sequencing.

4. Please include details of data analysis methodology in your manuscript Methods.

5. Please provide a sample size and power calculation in the Methods, or discuss the reasons for not performing one before study initiation.

6. In your Methods section, please provide additional information about the participant recruitment method and the demographic details of your participants. Please ensure you have provided sufficient details to replicate the analyses such as: a) the recruitment date range (month and year), b) a description of any inclusion/exclusion criteria that were applied to participant recruitment, d) a statement as to whether your sample can be considered representative of a larger population, d) a description of how participants were recruited, and e) descriptions of where participants were recruited and where the research took place.

7. Please ensure you have discussed any potential limitations of your study in the Discussion, including study design, sample size and/or potential confounders.

8. Please amend the manuscript submission data (via Edit Submission) to include author Eunae Sandra Cho,, Dawool Han, Jung Ah Park, Ju Yeong Park, Woong Nam, Hyung Jun Kim, In-Ho Cha.

9. Your ethics statement should only appear in the Methods section of your manuscript. If your ethics statement is written in any section besides the Methods, please delete it from any other section.

10. Please include captions for your Supporting Information files at the end of your manuscript, and update any in-text citations to match accordingly. Please see our Supporting Information guidelines for more information: http://journals.plos.org/plosone/s/supporting-information.

Reviewers' comments:

Reviewer's Responses to Questions

**Comments to the Author**

1. Is the manuscript technically sound, and do the data support the conclusions?

Reviewer #1: Partly

Reviewer #2: Yes

2. Has the statistical analysis been performed appropriately and rigorously? 

Reviewer #1: I Don't Know

Reviewer #2: Yes

3. Have the authors made all data underlying the findings in their manuscript fully available?

Reviewer #1: No

Reviewer #2: Yes

4. Is the manuscript presented in an intelligible fashion and written in standard English?

Reviewer #1: Yes

Reviewer #2: Yes

5. Review Comments to the Author

Reviewer #1: Thank for giving me an opportunity to review your interesting study.

I think that your study is so important. But there are some serious concerns as follows:

• First of all, although recurrence group includes some extensive cases (such as T4b case), I have a serious concern about the high rate of recurrence and positive margin. Please add the detailed surgical procedures (It is possible as supplemental information).

• I think that LB003 is not cT2.

• Please add the information of NM stage.

• LB001 is young patient. Why was adjuvant chemotherapy not performed?

• How many patients with oral cancer are treated in your hospital? Please add the detailed criteria of patient enrollment. How about patient selection bias?

I have ethical concerns in this study.

Reviewer #2: The manuscript describes a novel approach of using liquid biopsy for surveillance of patients with oral cancers. The method used is appropriate and well detailed. Though the sample collection part of the methodology could be written with more clarity by creating subsections for the different samples obtained.

The results was well expressed addressing the objectives of the research.

Authors state in the discussion (before limitation statement) that detection of ctDNA was more efficient using plasma than saliva in recurrence cases but stated in the result that saliva provides early detection which wanes in T4 stage of sample collection where plasma values increase. This statement needs to be modified.

In the limitation of the study, authors claim patients were in incurable state but had patients with T2 staged cancer which can be controlled or cured in many cases. The statement needs to be modified.

The limitation on the discussion should commence on a new paragraph.

Overall, it is a well-detailed and drafted manuscript.

6. PLOS authors have the option to publish the peer review history of their article (what does this mean?). If published, this will include your full peer review and any attached files.

Reviewer #1: **Yes: **Masaya Akashi

Reviewer #2: No

---

## [Author Response · Author response to Decision Letter 0]

14 Jun 2021

Point to Point response to Reviewer’s comments

We thank the Reviewers for their critical comments and helpful suggestions, which we addressed in full to improve the impact of our data and the quality of our manuscript. As described in detail below, we have revised our manuscript in response to the Reviewers’ comments. Below is a point-to-point response to the individual comments.

Editor (Remarks to the Author) 

1. Please ensure that your manuscript meets PLOS ONE's style requirements, including those for file naming. The PLOS ONE style templates can be found at https://journals.plos.org/plosone/s/file?id=wjVg/PLOSOne_formatting_sample_main_body.pdf

(Author’s Response) The format of the title page was changed according to the PLOSONE guidelines. 

2. Issues about competing interest of three authors (Hae-Suk Kim, Jung Ah Park, Ju Yeong Park) and role of funder. 

(Author’s Response) I would like to declare no competing interests exist, related to financial or non-financial (such as receiving funds, stocks, or shares), with Theragen Bio. Funding for conducting this study solely came from my own governmental grant written in acknowledgement (National Research Foundation of Korea (NRF) grant funded by the Korean Government). Hae-Suk Kim, Juyoung Park, and Jeonga Park, who are employees of Theragen Bio, solely contribute their scientific expertise and knowledge to conduct next-generation sequencing and bioinformatics analysis for this study. They do not have any competing interests. Also, I would like to declare that Theragen Bio did not provide any funding for this study and does not alter my adherence to PLOS ONE policies on sharing data and materials 

3. We note that you are reporting an analysis of a microarray, next-generation sequencing, or deep sequencing data set. PLOS requires that authors comply with field-specific standards for preparation, recording, and deposition of data in repositories appropriate to their field. Please upload these data to a stable, public repository (such as ArrayExpress, Gene Expression Omnibus (GEO), DNA Data Bank of Japan (DDBJ), NCBI GenBank, NCBI Sequence Read Archive, or EMBL Nucleotide Sequence Database (ENA)). In your revised cover letter, please provide the relevant accession numbers that may be used to access these data. For a full list of recommended repositories

(Author’s Response) Following your recommendation, next generation sequencing data were uploaded on NCBI SRA with accession number PRJNA718901 (https://www.ncbi.nlm.nih.gov/bioproject/PRJNA718901/)

4. Please include details of data analysis methodology in your manuscript Methods.

(Author’s Response) We included details of data analysis methodology regarding sample collection, sequencing data analysis and data curation in Methods. 

5. Please provide a sample size and power calculation in the Methods, or discuss the reasons for not performing one before study initiation.

(Author’s Response) We recorded the limitation and the reason of small sample size of the study as well as optimal sample size for repeated analysis in Discussion section. Altough Wang et al. performed liquid biopsy on 93 patients, repeated samples were only 9 patients within them with median follow up period 12 months after surgery [1].

Please consider that our study tried the earlier detection of circulating tumor DNA from 1 month to 6 months after sugery within 18 months clinical follow up. 

6. In your Methods section, please provide additional information about the participant recruitment method and the demographic details of your participants. Please ensure you have provided sufficient details to replicate the analyses such as: a) the recruitment date range (month and year), b) a description of any inclusion/exclusion criteria that were applied to participant recruitment, c) a statement as to whether your sample can be considered representative of a larger population, d) a description of how participants were recruited, and e) descriptions of where participants were recruited and where the research took place.

(Author’s Response) We added detailed sample information about the recruitment range, inclusion/exclusion criteria, power and limitation of sample size (in Discussion), recruitment methods in Method section.

7. Please ensure you have discussed any potential limitations of your study in the Discussion, including study design, sample size and/or potential confounders.

(Author’s Response) We revised the manuscript in Discussion section with potential limitations of the study. As the reviewers pointed out, small sample size would be the weakest point of our study. However, longitudinal, serial liquid sample collection as well as paired solid tumor analysis were strong point of our study.

8. Please amend the manuscript submission data (via Edit Submission) to include author Eunae Sandra Cho,, Dawool Han, Jung Ah Park, Ju Yeong Park, Woong Nam, Hyung Jun Kim, In-Ho Cha.

(Author’s Response) We revised the manuscript with detailed author contribution following the guidelines of your journal. Also all authors were included during resubmission.

9. Your ethics statement should only appear in the Methods section of your manuscript. If your ethics statement is written in any section besides the Methods, please delete it from any other section.

(Author’s Response) Following the guidelines, ethics statement was written only in Methods section. 

10. Please include captions for your Supporting Information files at the end of your manuscript, and update any in-text citations to match accordingly. Please see our Supporting Information guidelines for more information:

http://journals.plos.org/plosone/s/supporting-information.

(Author’s Response) We revised the manuscript following your journal guidelines.

Reviewer #1

1. First of all, although recurrence group includes some extensive cases (such as T4b case), I have a serious concern about the high rate of recurrence and positive margin. Please add the detailed surgical procedures (It is possible as supplemental information).

(Author's response) The limitations of this study in the discussion are the small sample size and the representativeness of the entire patient group. According to another study conducted in our institution, of 255 SCC patients, Stage I was 27.8%, Stage II 19.6%, Stage III 7.8%, Stage IVA 37.6%, Stage IVB 7.1%, in summary 47.4% for early stage and 52.6% for advanced stage [2]. In this study, the early stage was 36.4% and the advanced stage was 63.6%. LB003 contributed to a higher recurrence rate as this patient was revised to Stage IVB following reviewer's beneficial suggestion.

Summarizing the surgical procedure in relation to the positive margin, after a wide resection of the main mass during the operation, fresh-frozen biopsy was routinely performed at 5 to 7 sites in accordance with insurance criteria, and repeated until all are determined free. In the final pathologic diagnosis, the area that determines the margin cannot be completely consistent with the area where a fresh frozen biopsy was performed. In this reason, close or positive margin is reported in final pathologic diagnosis contrary to intraoperative frozen biopsy. Among the patients included in this study, patients with advanced stage had orbital, ethmoid, and pterygoid extensions where clear margin could not be obtained easily.

2. I think that LB003 is not cT2.

 (Author's response) The length of the long axis measured by preoperative MRI of LB003 patient was 3.8cm, but the adjacent muscles were not considered. In fact, the medical oncologist recorded it as Stage II, and the radiation oncologist doctor recorded as Stage IVB. After careful reviewing all participants, patient LB003 was revised to Stage IVB following reviewer’s suggestion.

3. Please add the information of NM stage

(Author's response) As requested by reviewer, we have added the NM stage in Table 1.

4. LB001 is young patient. Why was adjuvant chemotherapy not performed?

(Author's response) Even if CCRT is recommended by the Oral and Maxillofacial Surgeon following NCCN guidelines, radiation alone is sometimes administered in case of the patient refuses chemotherapy by personal reason after an individual interview with medical oncologist or radiation oncologist. In this case, the patient refused platinum based chemotherapy owing to low body weight and severe anorexia. 

5. How many patients with oral cancer are treated in your hospital? Please add the detailed criteria of patient enrollment. How about patient selection bias?

(Author's response) From 2000 to 2017, the total number of SCC patients with oral cancer at the research institute was 444. Among them, 255 patients with complete medical records were considered sufficient for academic study. Recently, about 40 SCC surgeries are performed each year, and sarcoma, mucosal melanoma, and salivary gland tumors are not included in this number. We have added detailed criteria in the Methods section. Although there is an opportunity of bias due to small sample size, more patients' participation in the study was not possible due to budget overruns and time limits. 

Reviewer #2:

1. Authors state in the discussion (before limitation statement) that detection of ctDNA was more efficient using plasma than saliva in recurrence cases but stated in the result that saliva provides early detection which wanes in T4 stage of sample collection where plasma values increase. This statement needs to be modified.

(Author's response) We revised the discussion section as review’s clear suggestion. As it has not yet been determined which liquid biopsy method should be applied as a standard for head and neck cancer (low-coverage whole genome sequencing, ctDNA targeted deep sequencing or HPV detection etc.), we wanted to note that the timing of sampling is clinically more important than the method.

2. In the limitation of the study, authors claim patients were in incurable state but had patients with T2 staged cancer which can be controlled or cured in many cases. The statement needs to be modified.

(Author's response) We revised that sentence to avoid confusion.

3. The limitation on the discussion should commence on a new paragraph.

(Author's response) As reviewer’s suggestion, we revised the limitation of study in discussion section on a paragraph. 

References

1. Wang Y, Springer S, Mulvey CL, Silliman N, Schaefer J, Sausen M, et al. Detection of somatic mutations and HPV in the saliva and plasma of patients with head and neck squamous cell carcinomas. Sci Transl Med. 2015;7(293):293ra104. Epub 2015/06/26. doi: 10.1126/scitranslmed.aaa8507. PubMed PMID: 26109104; PubMed Central PMCID: PMCPMC4587492.

2. Kim DW, Lee S, Kwon S, Nam W, Cha IH, Kim HJ. Deep learning-based survival prediction of oral cancer patients. Sci Rep. 2019;9(1):6994. Epub 2019/05/08. doi: 10.1038/s41598-019-43372-7. PubMed PMID: 31061433; PubMed Central PMCID: PMCPMC6502856.

---

## [Decision Letter · Decision Letter 1]

30 Jun 2021

PONE-D-21-08145R1

Longitudinal detection of somatic mutations in saliva and plasma for the surveillance of oral squamous cell carcinomas

PLOS ONE

Dear Dr. Cha Yong Hoon,

Thank you for submitting your manuscript to PLOS ONE. After careful consideration, we feel that it has merit but does not fully meet PLOS ONE’s publication criteria as it currently stands. Therefore, we invite you to submit a revised version of the manuscript that addresses the points raised during the review process.

ACADEMIC EDITOR:

Please revise your manuscript further. 

We look forward to receiving your revised manuscript.

Kind regards,

Zhaoqiang Zhang

Academic Editor

PLOS ONE

Journal Requirements:

Reviewers' comments:

Reviewer's Responses to Questions

**Comments to the Author**

1. If the authors have adequately addressed your comments raised in a previous round of review and you feel that this manuscript is now acceptable for publication, you may indicate that here to bypass the “Comments to the Author” section, enter your conflict of interest statement in the “Confidential to Editor” section, and submit your "Accept" recommendation.

Reviewer #1: All comments have been addressed

Reviewer #2: All comments have been addressed

2. Is the manuscript technically sound, and do the data support the conclusions?

Reviewer #1: Partly

Reviewer #2: Yes

3. Has the statistical analysis been performed appropriately and rigorously? 

Reviewer #1: I Don't Know

Reviewer #2: Yes

4. Have the authors made all data underlying the findings in their manuscript fully available?

Reviewer #1: Yes

Reviewer #2: Yes

5. Is the manuscript presented in an intelligible fashion and written in standard English?

Reviewer #1: Yes

Reviewer #2: Yes

6. Review Comments to the Author

Reviewer #1: Why was the number ctDNA detection different among patients (Figure 3c)? The authors mention that The presence of ctDNA in liquid biopsy indicated a substantial increased possibility of recurrence. Earlier ctDNA detection indicates the need for close clinical monitoring in the section of Results.

Please explain in more details.

Reviewer #2: Authors have addressed all the issues raised in the review. They have made adjustments to the text as indicated

7. PLOS authors have the option to publish the peer review history of their article (what does this mean?). If published, this will include your full peer review and any attached files.

Reviewer #1: No

Reviewer #2: No

---

## [Author Response · Author response to Decision Letter 1]

13 Aug 2021

Thank you for the thoughtful comments and suggestions. We revised the manuscript as following.

1. If the authors have adequately addressed your comments raised in a previous round of review and you feel that this manuscript is now acceptable for publication, you may indicate that here to bypass the “Comments to the Author” section, enter your conflict of interest statement in the “Confidential to Editor” section, and submit your "Accept" recommendation.

Reviewer #1: All comments have been addressed.

Reviewer #2: All comments have been addressed.

Thank you for all your confirmation.

2. Is the manuscript technically sound, and do the data support the conclusion?

Reviewer #1: Partly

Reviewer #2: Yes

3. Has the statistical analysis been performed appropriately and rigorously?

Reviewer #1: I don’t know

We agree with the reviewer. The statistical analysis is important to support for our results from the cohort. Pragmatically, we showed that ctDNA was detected approximately 4 months earlier than clinical recurrence in plasma or saliva of 3 out 6 recurred patients (Table 2 in manuscripts). Although the limitation of cohort size and sampling issue was discussed in discussion page 14, we would like to show whether variants are statistically detected in T2 and T3 which are earlier time point than clinical recurrence observation in our cohort. From this point of view, we used mean allele frequency of mutations detected in each patient at both T2 (1 month after surgery) and T3 (3 month after surgery). As you can see figure S1. below, the recurred group showed significantly positive detection of variant (mean AF, 0.05% for non-recurred versus 1.58% for recurred; median AF, 0% for non-recurred versus 0.6% for recurred; Wilcoxon signed rank test, P = 0.03).

It is unclear whether liquid biopsy could be useful in a real practice, we deemed that the result from this small cohort can be meaningful to evaluate the clinical feasibility of liquid biopsy for oral cancer recurrence early detection. Based on current results, we will have a plan to perform a clinical validation study with our branch hospitals. Therefore, we hope that our results and the new statistical analysis using T2 and T3 data to helpful for our further study design.

Reviewer #2: Yes

Thank you for your comments.

4. Have the authors made all data underlysing the findings in their manuscript fully available?

Reviewer #1: Yes

Reviewer #2: Yes

5. Is the manuscript presented in an intelligible fashion and written in standard English?

Reviewer #1: Yes

Reviewer #2: Yes

6. Review comments to the Author

Reviewer #1: Why the number ctDNA detection different among patients (Figure 3c)? The authors mention that the presence of ctDNA in liquid biopsy indicated a substantial increased possibility of recurrence. Earlier ctDNA detection indicates the need for close clinical monitoring in the section of results. Please explain in more details

Response: 

Thank you for your comments. We apologize that we did not clearly explained the figure. The purple triangle in Figure 3C indicates only ctDNA detection in either plasma or saliva during the study period. We attributed the missing samples to missed patient clinic visit and unmet the DNA quality to conduct the experiment in the results page 10 (Table 2, S4 table). And, the legend of figure 3C was revised.

And, we revised and added more data (Figure S1) to support the earlier ctDNA detection indicates the need for close clinical monitoring in the section of results page 11. 

Reviewer #2: Yes

---

## [Editor Report · Decision Letter 2]

20 Aug 2021

Longitudinal detection of somatic mutations in saliva and plasma for the surveillance of oral squamous cell carcinomas

PONE-D-21-08145R2

Dear Dr. Cha Yong Hoon,

We’re pleased to inform you that your manuscript has been judged scientifically suitable for publication and will be formally accepted for publication once it meets all outstanding technical requirements.

Kind regards,

Zhaoqiang Zhang

Academic Editor

PLOS ONE
---

## [Editor Report · Acceptance letter]

26 Aug 2021

PONE-D-21-08145R2 

Longitudinal detection of somatic mutations in saliva and plasma for the surveillance of oral squamous cell carcinomas

Dear Dr. Cha:

I'm pleased to inform you that your manuscript has been deemed suitable for publication in PLOS ONE. Congratulations! Your manuscript is now with our production department. 

Kind regards, 

on behalf of

Dr. Zhaoqiang Zhang 

Academic Editor

PLOS ONE